# Two distinct mechanisms target the autophagy-related E3 complex to the pre-autophagosomal structure

**Kumi Harada[1][†], Tetsuya Kotani[1][†], Hiromi Kirisako[1], Machiko Sakoh-Nakatogawa[1], Yu Oikawa[2], Yayoi Kimura[3], Hisashi Hirano[3], Hayashi Yamamoto[‡], Yoshinori Ohsumi, Hitoshi Nakatogawa[1]\***

[1]School of Life Science and Technology, Tokyo Institute of Technology, Yokohama, Japan; [2]Institute of Innovative Research, Tokyo Institute of Technology, Yokohama, Japan; [3]Advanced Medical Research Center, Yokohama City University, Yokohama, Japan

**Abstract** In autophagy, Atg proteins organize the pre-autophagosomal structure (PAS) to initiate autophagosome formation. Previous studies in yeast revealed that the autophagy-related E3 complex Atg12-Atg5-Atg16 is recruited to the PAS via Atg16 interaction with Atg21, which binds phosphatidylinositol 3-phosphate (PI3P) produced at the PAS, to stimulate conjugation of the ubiquitin-like protein Atg8 to phosphatidylethanolamine. Here, we discover a novel mechanism for the PAS targeting of Atg12-Atg5-Atg16, which is mediated by the interaction of Atg12 with the Atg1 kinase complex that serves as a scaffold for PAS organization. While autophagy is partially defective without one of these mechanisms, cells lacking both completely lose the PAS localization of Atg12-Atg5-Atg16 and show no autophagic activity. As with the PI3P-dependent mechanism, Atg12-Atg5-Atg16 recruited via the Atg12-dependent mechanism stimulates Atg8 lipidation, but also has the specific function of facilitating PAS scaffold assembly. Thus, this study significantly advances our understanding of the nucleation step in autophagosome formation.
DOI: https://doi.org/10.7554/eLife.43088.001

**\*For correspondence:**
hnakatogawa@bio.titech.ac.jp

[†]These authors contributed equally to this work

**Present address:** [‡]Graduate School and Faculty of Medicine, The University of Tokyo, Tokyo, Japan

## Introduction

Macroautophagy (hereafter autophagy) is a major route for transport of intracellular material into lysosomes or vacuoles in almost all eukaryotes (*Ohsumi, 2014*; *Yang and Klionsky, 2010*). In autophagy, a membrane cisterna called the isolation membrane (or phagophore) is generated, expands, becomes spherical, and closes to form a double membrane vesicle called the autophagosome. During the course of this process, various cytoplasmic components, including proteins, RNA, and organelles, are selectively or non-selectively sequestered into the autophagosome. The autophagosome fuses with the lysosome/vacuole to allow degradation of the contents. An increasing number of studies have suggested that autophagy is involved in the regulation of a wide range of cellular functions, and linked to a variety of human diseases (*Bento et al., 2016*; *Dikic and Elazar, 2018*; *Mizushima, 2018*).

Isolation of autophagy-defective mutants of the budding yeast *Saccharomyces cerevisiae* and subsequent analysis of these mutants led to the identification of autophagy-related (*ATG*/Atg) genes/proteins. Among the over 40 Atg proteins that have been identified to date, 19 are directly involved in the biogenesis of the autophagosome induced under starvation (*Nakatogawa et al., 2009*; *Ohsumi, 2014*; *Yang and Klionsky, 2010*). These 'core' Atg proteins constitute six functional units: (i) the Atg1 kinase complex; (ii) Atg9 vesicles; (iii) phosphatidylinositol (PI) 3-kinase (PI3K) complex I ; (iv) the Atg2-Atg18 complex; (v) the Atg12 conjugation system; and (vi) the Atg8

conjugation system. In response to starvation, these proteins interact with each other, localize to the site of autophagosome formation in an ordered manner, and organize the pre-autophagosomal structure (PAS) (**Nakatogawa et al., 2009**; **Suzuki et al., 2001**; **Suzuki and Ohsumi, 2010**), in which a precursor of the autophagosomal membrane is generated. The molecular basis of PAS organization, including how Atg proteins are recruited to the PAS, is a key question that needs to be addressed to understand the 'nucleation' step in autophagosome formation.

The ubiquitin-like protein Atg12 is covalently attached to a lysine residue in Atg5 via ubiquitin-like conjugation reactions, resulting in the Atg12-Atg5 conjugate (**Mizushima et al., 1998a**; **Mizushima et al., 1998b**). Atg12-Atg5 non-covalently interacts with Atg16 (Atg16L in mammals) to form the Atg12-Atg5-Atg16/Atg16L complex (hereafter the Atg16/Atg16L complex) (**Kuma et al., 2002**; **Mizushima et al., 2003**; **Mizushima et al., 1999**). The Atg16/Atg16L complex is localized to the PAS (or the site of autophagosome formation) and acts as an E3 enzyme to stimulate the conjugation reaction of ubiquitin-like Atg8/LC3-family proteins to the lipid phosphatidylethanolamine (PE) in autophagosome intermediates (i.e., a still-unknown membrane component of the PAS and the isolation membrane) (**Suzuki et al., 2007**; **Hanada et al., 2007**; **Ichimura et al., 2000**; **Fujita et al., 2008**; **Nakatogawa, 2013**). Atg8-PE conjugates promote the expansion of the isolation membrane (**Nakatogawa et al., 2007**; **Xie et al., 2008**), and also bind to autophagy receptors that recognize specific degradation targets for their selective sequestration into the autophagosome (**Gatica et al., 2018**).

In both yeast and mammals, the recruitment of the Atg16/Atg16L complex to the site of autophagosome formation depends on PI3-phosphate (PI3P) produced by PI3K complex I (**Itakura and Mizushima, 2010**; **Suzuki et al., 2007**). A recent study in mammalian cells revealed that the PROPPIN family protein WIPI2b binds both Atg16L1 and PI3P to target the Atg16L1 complex to autophagosome formation sites (**Dooley et al., 2014**). In *S. cerevisiae*, Atg21, one of the three WIPI homologs, was shown to mediate this process in a similar manner (**Juris et al., 2015**). However, knockout of *ATG21* did not completely abrogate the PAS localization of the Atg16 complex or the autophagic activity of cells (**Meiling-Wesse et al., 2004**; **Nair et al., 2010**; **Strømhaug et al., 2004**), suggesting that there is an unknown mechanism which directs the Atg16 complex to the PAS, in addition to the PI3K complex I -PI3P-Atg21 axis.

In this study, we identified the Atg1 kinase complex, which forms a scaffold for PAS organization, as a novel interaction partner of the Atg16 complex, and found that this intercomplex interaction collaborates with the PI3P-dependent mechanism to recruit the Atg16 complex to the PAS. In addition to the stimulation of Atg8 lipidation, the Atg16 complex recruited via this newly discovered mechanism has a specific, non-E3 function: the promotion of PAS scaffold assembly.

## Results

### An Atg12-dependent, PI3P-independent mechanism targets the Atg16 complex to the PAS

To clarify the mechanism that directs the Atg16 complex to the PAS, we carefully analyzed the PAS localization of this complex in cells lacking different Atg proteins (**Figure 1**). In this analysis, the Atg16 complex was visualized by Atg5-GFP, and the PAS was labeled with the scaffold protein Atg17 fused with mCherry (**Suzuki et al., 2007**). In the currently accepted model, Atg5 and Atg16 cooperate to target the complex to the PAS, whereas Atg12 is dispensable for this process (**Suzuki et al., 2007**). It is also believed that PI3P produced by PI3K complex I , which contains Atg14 as a specific subunit, is essential for the localization of the Atg16 complex to the PAS. This PI3P-dependency could, at least in part, be explained by the role of the PI3P-binding protein Atg21 that interacts with Atg16 to recruit the Atg16 complex to the PAS (**Nair et al., 2010**; **Juris et al., 2015**). In agreement with this model, the PAS localization of Atg5 was lost by deletion of *ATG16* (**Figure 1**). It was also confirmed that Atg5 localized to the PAS less efficiently in the absence of Atg21. Deletion of *ATG14* decreased the PAS localization of Atg5; however, Atg5 still significantly localized to the PAS even without Atg14, to an extent similar to that observed in cells lacking Atg21. In addition, we noticed that the frequency of Atg5 localization to the PAS was decreased in the absence of Atg12, although it abnormally accumulated at the PAS. We found that PAS localization of Atg5 was totally abolished in cells lacking both Atg14 and Atg12 (**Figure 1**). Disruption of *ATG12*

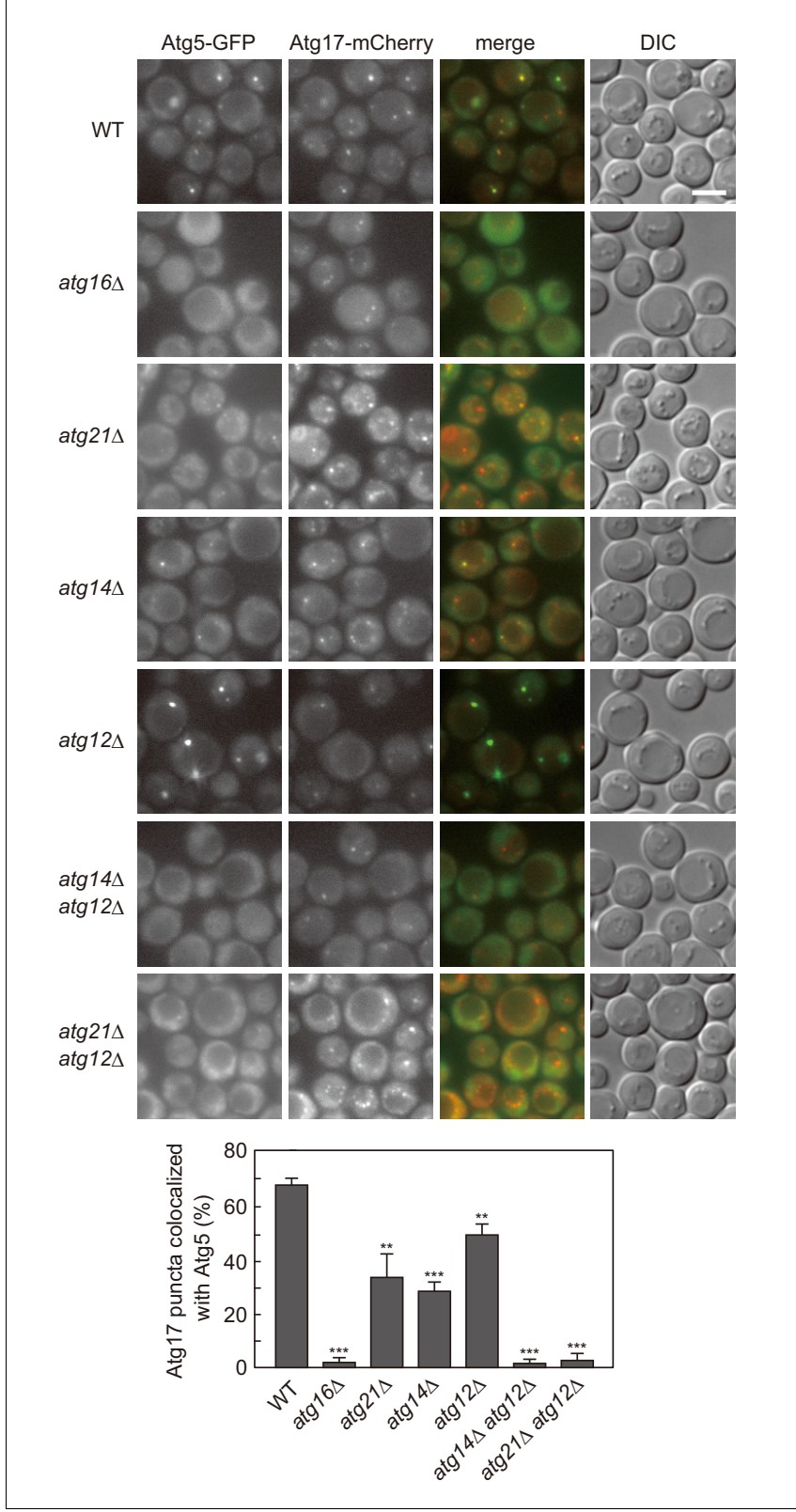

**Figure 1.** Atg12- and PI3P-dependent mechanisms cooperatively act to recruit the Atg16 complex to the PAS. Cells expressing Atg5-GFP and Atg17-mCherry were treated with rapamycin for 90 min, and analyzed by fluorescence microscopy. DIC, Differential interference contrast microscopy. Bars, 5 μm. The ratios of Atg17-

*Figure 1 continued on next page*

*Figure 1 continued*

mCherry puncta positive for Atg5-GFP to total Atg17-mCherry puncta were calculated, and the mean values are shown with standard deviations (n = 3). **p<0.01; ***p<0.001 (unpaired two-tailed Student's *t*-test).
DOI: https://doi.org/10.7554/eLife.43088.002

also abrogated the residual PAS localization of Atg5 in *ATG21* knockout cells. These results suggest that in addition to the previously described PI3P-dependent pathway, there exists an uncharacterized, PI3P-independent mechanism that targets the Atg16 complex to the PAS, which likely involves Atg12.

## The Atg16 complex interacts with the Atg1 complex under autophagy-inducing conditions

We proceeded to investigate an Atg12-dependent mechanism for PAS-targeting of the Atg16 complex. Yeast cells expressing FLAG-tagged Atg5 (Atg5-FLAG) were treated with rapamycin, which inhibits Tor kinase complex 1 and thereby induces various starvation responses including autophagy even in the presence of nutrients (*Noda and Ohsumi, 1998*), followed by immunoprecipitation using anti-FLAG antibody. Mass spectrometry of the immunoprecipitates identified a number of proteins as possible interaction partners of the Atg16 complex, and included most components of the Atg1 complex (*Figure 2—figure supplement 1A and B*). The Atg1 complex is composed of the protein kinase Atg1 and the regulatory and scaffold proteins Atg13, Atg17, Atg29, and Atg31, and triggers autophagosome formation in response to nutrient starvation (*Fujioka et al., 2014*; *Kamada et al., 2000*). Atg1, Atg17, and Atg29 could also be detected in Atg5-FLAG immunoprecipitates by immunoblotting (*Figure 2A* and *Figure 2—figure supplement 1C*). When *ATG12* or *ATG16* was deleted, coimmunoprecipitation of Atg17 with Atg5-FLAG was largely decreased (*Figure 2A* and *Figure 2—figure supplement 1B*). We also showed that Atg17 was not precipitated with Atg5-FLAG in the absence of Atg10, which is essential for Atg12 conjugation to Atg5 (*Mizushima et al., 1998a*) (*Figure 2A*). Immunoprecipitation of FLAG-tagged Atg16 also precipitated Atg17; however, this was lost by knockout of *ATG5* or *ATG12* (*Figure 2B*). Thus, the formation of the Atg16 complex is required for its interaction with the Atg1 complex. We also examined this interaction in cells lacking any of the components of the Atg1 kinase complex. In this analysis, coprecipitation of Atg1 with Atg5-FLAG was also examined to evaluate the effect of *ATG17* knockout on the association between the two complexes. The results clearly showed that all the components of the Atg1 complex are important for its interaction with the Atg16 complex (*Figure 2C*). In addition, the F430A mutation in Atg13, which impairs the formation of the Atg1 complex (*Yamamoto et al., 2016*), reduced Atg17 precipitation with the Atg16 complex (*Figure 2D*). These results suggest that the formation of the Atg1 complex is a prerequisite for its association with the Atg16 complex. In contrast, *ATG14* deletion did not affect Atg17 precipitation with Atg5-FLAG (*Figure 2E*), consistent with the idea that this novel intercomplex interaction is involved in the PI3P-independent PAS targeting of the Atg16 complex.

We found that the Atg16 complex interacts with the Atg1 complex depending on cell treatment with rapamycin (*Figure 2F*). Consistent with this result, the interaction was not considerably decreased by the absence of Atg11, which binds to the Atg1 complex but is dispensable for starvation-induced autophagy (*Kamada et al., 2000*; *Kim et al., 2001*). Upon nutrient starvation, Atg1, Atg13, and the Atg17-Atg29-Atg31 complex form the Atg1 complex, and multiple copies of the complex further associate with each other, leading to activation of Atg1 kinase via intermolecular autophosphorylation (*Yamamoto et al., 2016*; *Yeh et al., 2011*; *Yeh et al., 2010*). This assemblage of the Atg1 complexes serves as a scaffold to recruit downstream Atg proteins for PAS organization. The interaction between the Atg16 complex and the Atg1 complex was lost in the F375A mutant of Atg13 (*Figure 2D*), which can form the Atg1 complex, but is defective in its higher order assembly (*Yamamoto et al., 2016*). By contrast, the D211A mutation in Atg1, which abolishes its kinase activity (*Matsuura et al., 1997*), did not affect the interaction between the Atg16 and Atg1 complexes (*Figure 2G*). These results suggest that the Atg16 complex associates with the Atg1 complex following its supramolecular assembly in a manner independent of Atg1 kinase activity.

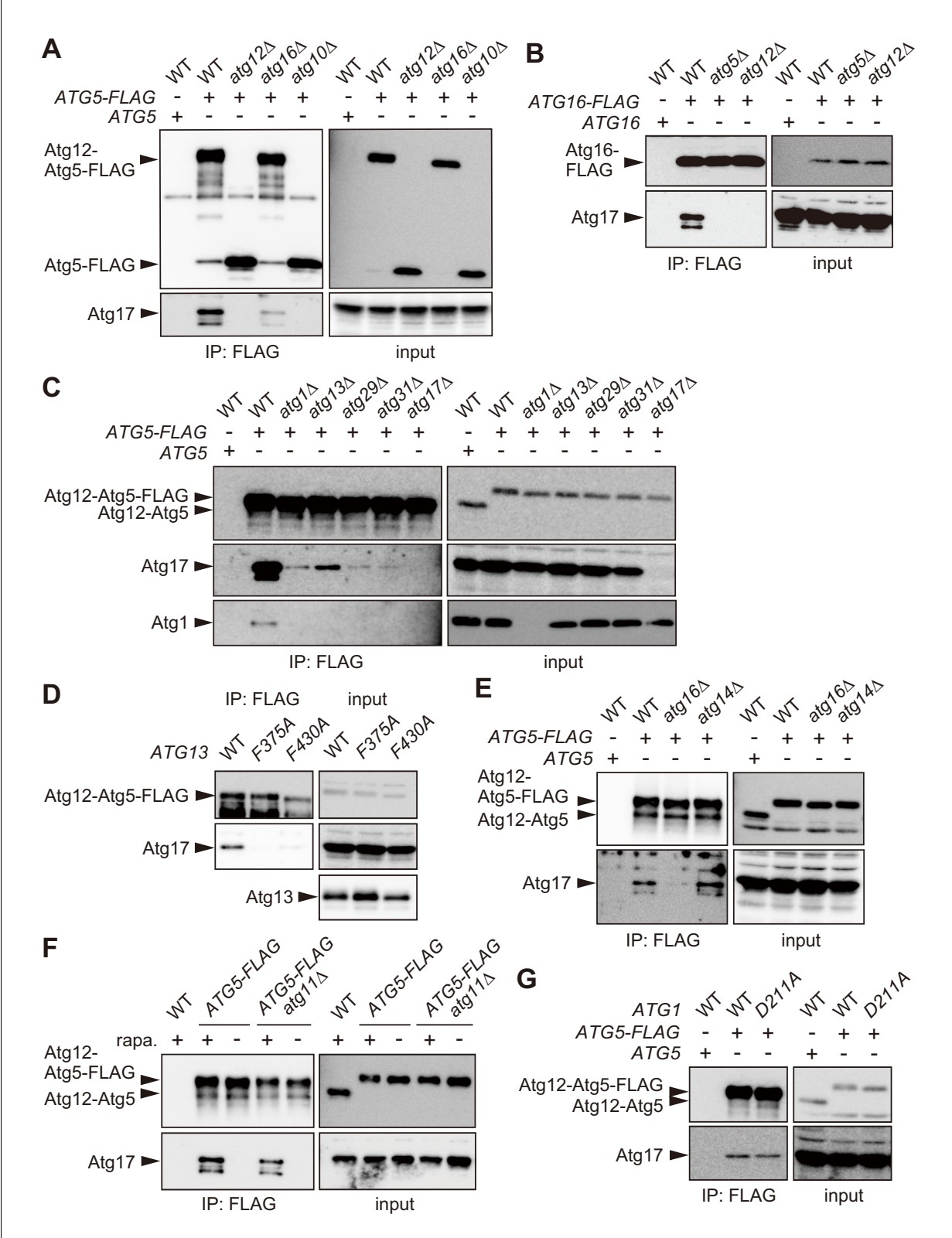

**Figure 2.** The Atg16 complex interacts with the Atg1 complex. (**A–C, E**) Yeast cells expressing Atg5-FLAG (**A, C–E**) or Atg16-FLAG (**B**) from each chromosomal locus were treated with rapamycin for 2 hr, and subjected to immunoprecipitation using anti-FLAG antibody. The immunoprecipitates were analyzed by immunoblotting using antibodies against FLAG (**A, B**), Atg12 (**C, E**), Atg17 (**A–C, E**), and Atg1 (**C**). (**D**) *atg13Δ* cells expressing wild-type Atg13, the F375A mutant, or the F430A mutant from centromeric plasmids were treated with rapamycin for 2 hr, subjected to immunoprecipitation

*Figure 2 continued on next page*

*Figure 2 continued*
using anti-FLAG antibody, and the immunoprecipitates were analyzed by immunoblotting using antibodies against Atg12, Atg13 and Atg17. (**F**) Yeast cells were treated with or without rapamycin for 2 hr, and coimmunoprecipitation of Atg17 with Atg5-FLAG was examined as described in *Figure 2C*. (**G**) Coimmunoprecipitation of Atg17 with Atg5-FLAG was analyzed in cells expressing wild-type Atg1 or the D211A mutant from the original chromosomal locus as described in *Figure 2C*.

DOI: https://doi.org/10.7554/eLife.43088.003

The following figure supplements are available for figure 2:

**Figure supplement 1.** Proteomic analysis to identify proteins bound to the Atg16 complex.

DOI: https://doi.org/10.7554/eLife.43088.004

**Figure supplement 2.** Atg17 interacts with Atg12.

DOI: https://doi.org/10.7554/eLife.43088.005

We also examined which subunits mediate the interaction between the Atg16 and Atg1 complexes. Yeast two-hybrid assay suggested that Atg12 could bind Atg17 and Atg31 (*Figure 2—figure supplement 2A*). In this assay, the Atg12-Atg17 interaction was still observed in *atg13Δ atg31Δ* cells (*Figure 2—figure supplement 2B*), in which Atg17 should not interact with the remaining subunits Atg1 and Atg29. By contrast, the Atg12-Atg31 interaction was abolished by *ATG17* knockout, suggesting that Atg12 interacted with Atg31 via Atg17 (*Figure 2—figure supplement 2C*). In addition, immunoprecipitation of Atg12 C-terminally fused with GFP, which is not associated with Atg5 and Atg16, coprecipitated Atg17 in cells lacking the other four subunits of the Atg1 complex, when both of Atg12-GFP and Atg17 were overexpressed. These results suggest that the interaction between Atg17 and Atg12 mediates the association of the two complexes.

## The Atg16 complex interacts with the Atg1 complex to localize to the PAS

Next, we examined the significance of the interaction between the Atg16 and Atg1 complexes in autophagosome formation. Atg12 is a ubiquitin-like protein with an approximately 100 amino acid-long extension at the N terminus (*Suzuki et al., 2005*). A previous study reported that while the ubiquitin-like domain of Atg12 was essential for autophagy, deletion of the N-terminal region caused a partial defect (*Hanada and Ohsumi, 2005*). The N-terminal region of Atg12 was not required for Atg12 conjugation to Atg5, the E3 activity of the conjugate, or the interaction of the conjugate with Atg16 (*Hanada and Ohsumi, 2005*) (*Figure 3—figure supplement 1*). Thus, the role for the Atg12 N-terminal region remained unknown. We found that the Atg1 complex was hardly coimmunoprecipitated with the Atg16 complex that contained Atg12 lacking the N-terminal 56 residues (Atg12$^{ΔN56}$) (*Figure 3A*), suggesting that the Atg12 N-terminal region is involved in the interaction of the Atg16 complex with the Atg1 complex. The results obtained by fluorescence microscopy (*Figure 1*) suggested that this interaction cooperates with the PI3P-dependent pathway in the recruitment of the Atg16 complex to the PAS. Therefore, we examined the PAS localization of the complex containing Atg12$^{ΔN56}$ in the absence of Atg21. While expression of wild-type Atg12 rescued a defect in the PAS localization of Atg5-GFP in *atg21Δ atg12Δ* cells, expression of the Atg12$^{ΔN56}$ mutant did not (*Figure 3B*). We also performed an alkaline phosphatase (ALP) assay to assess autophagic activity in the mutant cells. In this assay, a mutant form of the vacuolar phosphatase Pho8 (Pho8Δ60) is expressed in an unprocessed, inactive form in the cytoplasm. This mutant phosphatase is delivered into the vacuole through autophagy. Once inside the vacuole, it is processed into an active form, and its activity can be quantified biochemically (*Noda et al., 1995*). Consistent with previous results (*Hanada and Ohsumi, 2005*; *Meiling-Wesse et al., 2004*; *Strømhaug et al., 2004*), *atg12$^{ΔN56}$* cells (*atg12Δ* cells carrying the *atg12$^{ΔN56}$* plasmid) and *atg21Δ* cells (*atg21Δ atg12Δ* cells carrying the *ATG12$^{WT}$* plasmid) were only partially defective in autophagy (*Figure 3C*). When these mutations were combined (*atg21Δ atg12Δ* cells carrying the *atg12$^{ΔN56}$* plasmid), the cells showed almost no autophagic activity. These results suggest that the interaction of the Atg16 complex with the Atg1 complex indeed acts to target the Atg16 complex to the PAS, and that defects caused by the absence of this interaction can be partly compensated by the PI3P-dependent mechanism. If both of these mechanisms are simultaneously compromized, cells totally lose their ability to form the autophagosome.

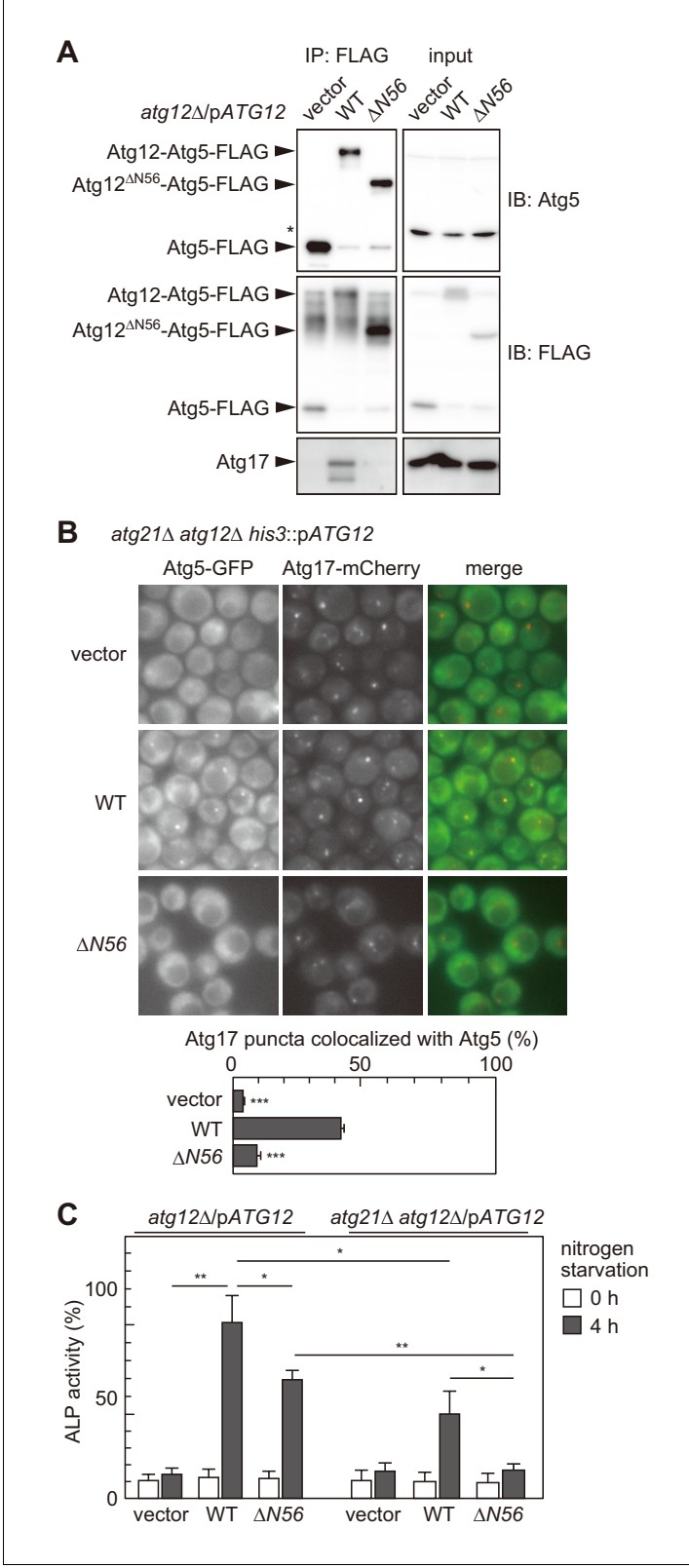

**Figure 3.** The interaction of the Atg16 complex with the Atg1 complex is involved in the PAS targeting of the Atg16 complex. (**A**) *atg12Δ* cells expressing wild-type Atg12 or Atg12$^{ΔN56}$ from centromeric plasmids were treated with rapamycin for 2 hr, and examined for coimmunoprecipitation of Atg17 with Atg5-FLAG as described in *Figure 2C*. The upper and middle panels were immunoblots obtained using antibodies against Atg5 and FLAG,

*Figure 3 continued on next page*

*Figure 3 continued*

respectively. Asterisk, non-specific bands. **(B)** Yeast cells were treated with rapamycin for 2 hr, and the PAS localization of Atg5-GFP was assessed by fluorescence microscopy as described in *Figure 1*. **p<0.01; ***p<0.001 (unpaired two-tailed Student's *t*-test). **(C)** *atg12Δ* and *atg12Δ atg21Δ* cells expressing wild-type Atg12 or Atg12$^{ΔN56}$ from centromeric plasmids were grown in nutrient-rich medium (open bars) and then starved in SD-N medium for 4 hr (closed bars), and their autophagic activities were evaluated by ALP assay. The mean values are shown with standard deviations (n = 3). *p<0.05; **p<0.01 (unpaired two-tailed Student's *t*-test).

DOI: https://doi.org/10.7554/eLife.43088.006

The following figure supplement is available for figure 3:

**Figure supplement 1.** The N-terminal region of Atg12 is not required for the E3 activity of the Atg12-Atg5 conjugate.

DOI: https://doi.org/10.7554/eLife.43088.007

## The Atg16 complex recruited via the Atg12-dependent pathway plays two different roles in PAS organization

Previous studies showed that PI3P-dependent PAS recruitment of the Atg16 complex is important for the production of Atg8-PE (*Meiling-Wesse et al., 2004*; *Strømhaug et al., 2004*). We asked whether the Atg12-dependent mechanism also contributes to this process. Atg8-PE production is stimulated upon nitrogen starvation (*Figure 4A*, *atg12Δ*/p*ATG12*$^{WT}$). As reported previously (*Meil-ing-Wesse et al., 2004*; *Strømhaug et al., 2004*), *ATG21* knockout significantly reduced the level of Atg8-PE, which still gradually increased during nitrogen starvation (*atg21Δ atg12Δ*/p*ATG12*$^{WT}$). We found that deletion of the N-terminal region of Atg12 also partially decreased Atg8-PE formation (*atg12Δ*/p*atg12*$^{ΔN56}$). In addition, starvation-induced Atg8-PE formation was totally abolished in cells lacking both Atg21 and the N-terminal region of Atg12 (*atg21Δ atg12Δ*/p*atg12*$^{ΔN56}$). Of note, the residual amount of Atg8-PE in these mutant cells should represent the conjugates that were pro-duced in the vacuolar membrane depending on the Atg16 complex, which is dispersed throughout the cytoplasm, independent of autophagy (*Nakatogawa et al., 2012a*). These results demonstrated that the Atg16 complex recruited by the Atg12-dependent mechanism acts as an E3 enzyme and promotes Atg8 lipidation, as was observed with the PI3P-dependent mechanism.

We noticed that Atg17-GFP puncta, which represent PAS scaffold assembly (*Yamamoto et al., 2016*), were decreased by knockout of *ATG12*, which disrupts the Atg12-dependent mechanism, whereas the puncta were increased by knockout of *ATG14* or *ATG21*, which impairs the PI3P-depen-dent mechanism (*Figure 1*). We confirmed this finding using *atg11Δ* cells, in which the starvation-induced assembly of the PAS scaffold can be assessed separately from a similar process that occurs under nutrient-replete conditions for the cytoplasm-to-vacuole targeting (Cvt) pathway (*Cheong et al., 2008*; *Kawamata et al., 2008*). The results demonstrated that the formation of Atg17-GFP puncta upon rapamycin treatment was defective in the absence of Atg12, Atg5, or Atg16 (*Figure 4B*). More importantly, Atg17-GFP puncta were also decreased by deletion of the N-terminal region of Atg12 (*Figure 4C*). These results suggest that the Atg16 complex recruited via the Atg12-dependent mechanism (i.e., the interaction with the Atg1 complex) facilitates starvation-induced PAS scaffold assembly.

## Discussion

Previous studies established a model for PAS targeting of the Atg16 complex: Atg16 interacts with Atg21, which binds PI3P produced by PI3K complex I at the PAS (*Juris et al., 2015*; *Nair et al., 2010*; *Meiling-Wesse et al., 2004*; *Strømhaug et al., 2004*) (*Figure 5*, PI3P-dependent targeting). This model nicely explained the PI3P-dependency of the process. However, disruption of *ATG21* or *ATG14* did not totally abolish PAS localization of the Atg16 complex, suggesting the existence of another pathway that targets this complex to the PAS in a PI3P-independent manner. In this study, we discovered that the Atg16 complex also interacts with the Atg1 complex via the N-terminal region of Atg12 to localize to the PAS (*Figure 5*, Atg12-dependent targeting). Thus, the Atg16 complex is recruited to the PAS through two different pathways. Although disrupting either of the path-ways caused partial defects, in the absence of both of these pathways, the Atg16 complex hardly localized to the PAS, and autophagy was completely blocked. These results suggested that these

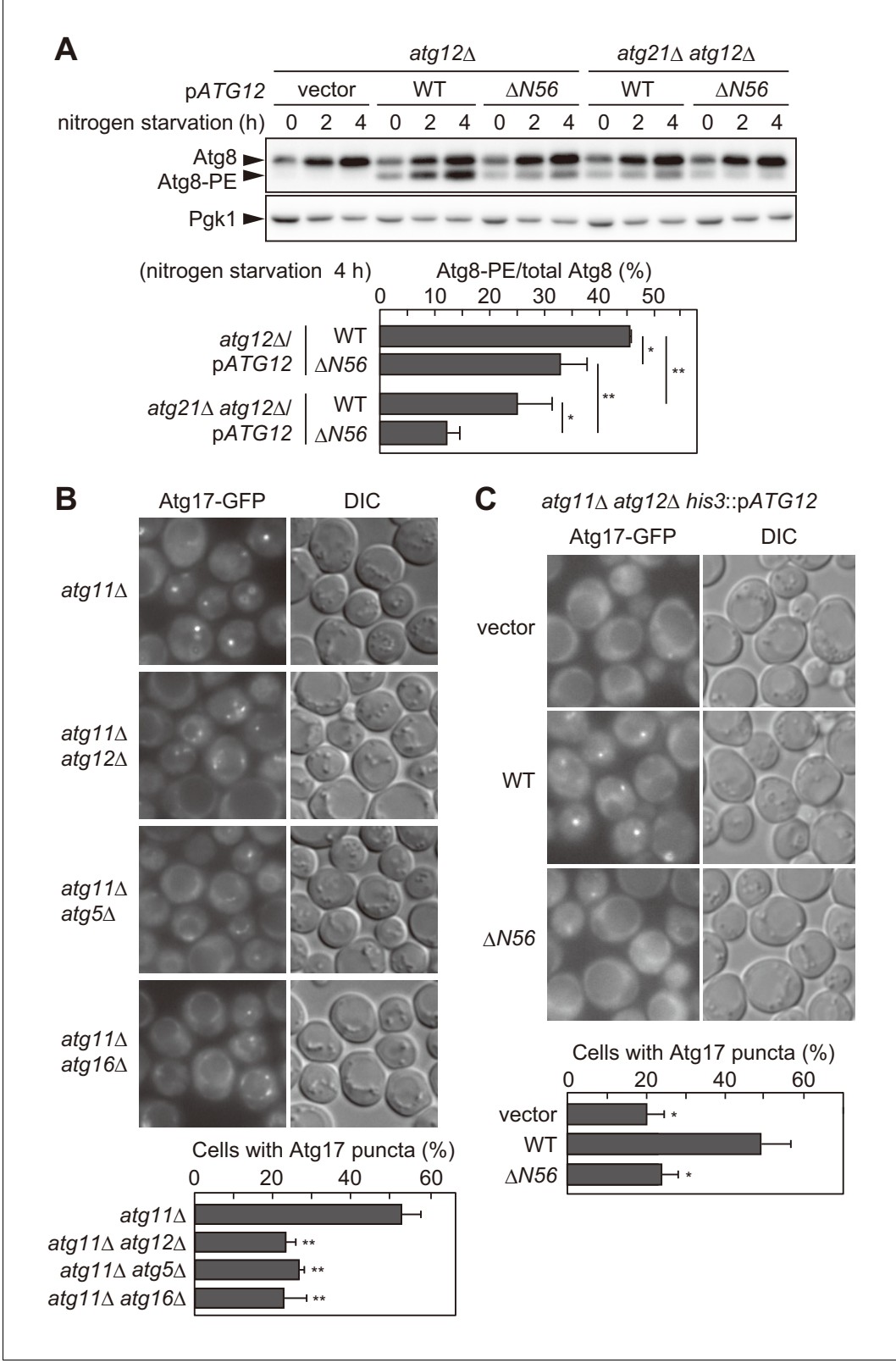

**Figure 4.** The Atg16 complex recruited via the association with the Atg1 complex facilitates Atg8 lipidation and PAS scaffold assembly. (**A**) Yeast cells were incubated in nitrogen starvation medium and examined for the production of Atg8-PE by urea-SDS-PAGE and immunoblotting using anti-Atg8 antibodies (see Materials and methods). The ratio of Atg8-PE to total Atg8 was calculated, and the mean values are shown with standard

*Figure 4 continued on next page*

*Figure 4 continued*

deviations (n = 3). *p<0.05; **p<0.01 (unpaired two-tailed Student's *t*-test). Pgk1 serves as a loading control. (**B and C**) Yeast cells expressing Atg17-GFP were treated with rapamycin for 90 min (**B**) or 2 hr (**C**), and observed under a fluorescence microscope. The proportion of cells containing Atg17-GFP puncta to total cells was calculated, and the mean values are shown with standard deviations (n = 3). **p<0.01 (unpaired two-tailed Student's *t*-test).

DOI: https://doi.org/10.7554/eLife.43088.008

pathways function in a partially redundant manner. Consistent with this idea, we showed that these pathways cooperatively act to stimulate Atg8-PE formation in response to starvation (*Figure 4A*). However, the Atg1 complex serves as a scaffold to initiate PAS organization, whereas Atg21 is recruited at a later step, following the production of PI3P by PI3K complex I (*Figure 5*), suggesting that there is a functional difference between these pathways. Indeed, we found that the Atg12-dependent pathway, but not the PI3P-dependent pathway, is involved in PAS scaffold assembly. Thus, the Atg16 complex recruited by the Atg12-dependent pathway has a specific, non-E3 role in the initiation of PAS organization. Given the complementary relationship with the PI3P-dependent pathway, it is likely that the Atg16 complex recruited through the association of the Atg1 complex also contributes to Atg8 lipidation at later stages (*Figure 5*, dashed arrow).

The interaction between the Atg16 complex and the Atg1 complex required both of the complexes to be intact. In addition, higher order assembly of Atg1 complexes was a prerequisite for the interaction (*Figure 2D*). We speculate that the Atg16 complex, in which Atg12-Atg5-Atg16 is dimerized by the homodimerization of Atg16 (*Fujioka et al., 2010*), simultaneously binds to two copies of the Atg1 complexes via the N-terminal regions of Atg12 (*Figure 5*). This mode of interaction can crosslink the Atg1 complexes, resulting in the facilitation of supramolecular assembly of Atg1 complexes to form the PAS scaffold.

In this study, we showed that the association between the Atg16 and Atg1 complexes requires the N-terminal region of Atg12. Although there is no sequence similarity between the N-terminal regions of yeast and mammalian Atg12, a recent study revealed that the Atg16L complex also

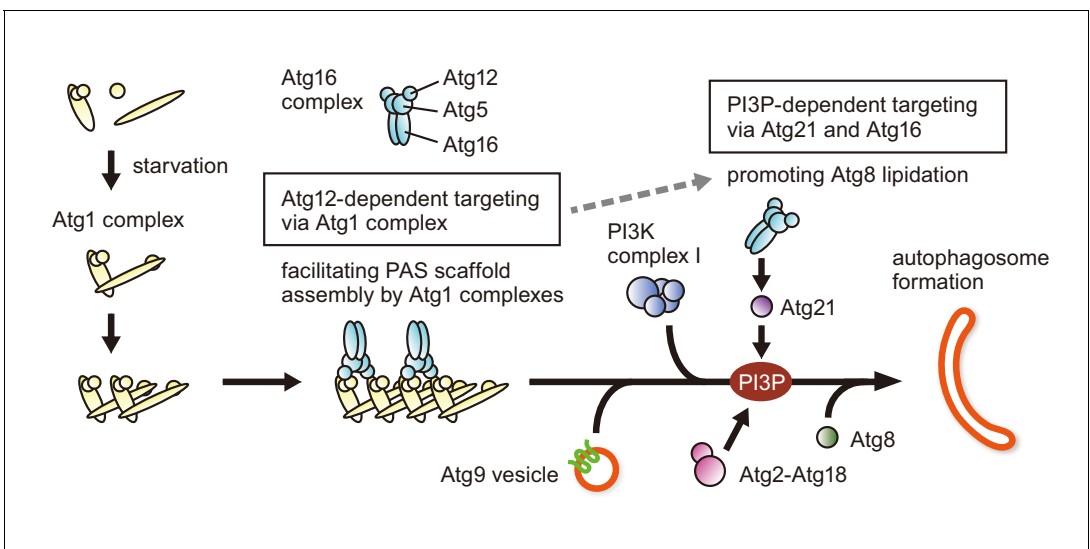

**Figure 5.** Model for the PAS recruitment of the Atg16 complex. The Atg16 complex is recruited to the PAS through two different pathways (Atg12-dependent targeting and PI3P-dependent targeting). Upon autophagy induction (starvation or TORC1 inactivation), the Atg1 complex is assembled, and multiple copies of Atg1 complexes further form a higher order assembly. During the process, the Atg16 complex associates with Atg1 complexes via the N-terminal region of Atg12, promoting PAS scaffold assembly. The Atg16 complex recruited at this stage also facilitates lipidation of Atg8 at a later stage in PAS organization (dashed arrow). As reported previously, following the recruitment of PI3K complex I and the production of PI3P by this complex, the Atg16 complex localizes to the PAS via the interaction with the PI3P-binding protein Atg21 to stimulate Atg8 lipidation.

DOI: https://doi.org/10.7554/eLife.43088.009

associates with the ULK1 complex (corresponding to the Atg1 complex) in mammalian cells (*Nishimura et al., 2013*). This intercomplex association was mediated by the interaction between Atg16L and FIP200, which is a mammalian counterpart of yeast Atg17, and important for the localization of the Atg16L complex to the isolation membrane. Thus, although the underlying mechanisms are different, the recruitment of the Atg16/Atg16L complex to autophagosome intermediates through association with the Atg1/ULK1 complex is a common process during autophagosome formation in yeast and mammals. It is still unclear how this process cooperates with the PI3P (WIPI2)-dependent mechanism in mammalian cells. The Atg16L complex may also facilitate supramolecular assembly of ULK complexes in the initiation of autophagosome formation.

## Materials and methods

### Yeast strains and media

*S. cerevisiae* strains used in this study are listed in *Table 1*. Gene knockout and tagging were performed as described previously (*Janke et al., 2004*). Yeast cells were grown at 30°C in YPD medium (1% yeast extract, 2% peptone, and 2% glucose) for immunoprecipitation analysis or in SD+CA+ATU medium [0.17% yeast nitrogen base without amino acids and ammonium sulfate (YNB w/o aa and as), 0.5% ammonium sulfate, 0.5% casamino acids, and 2% glucose supplemented with 0.002% adenine sulfate, 0.002% tryptophan, and 0.002% uracil] for fluorescence microscopy. Cells carrying pRS316-derived plasmids expressing Atg12, Atg13, and their mutants were cultured in SD+CA+ATU without uracil. To induce autophagy, cells were treated with 0.2 µg/mL rapamycin or incubated in SD-N medium (0.17% YNB w/o aa and as and 2% glucose).

### Plasmids

pRS316-based centromeric plasmids for expression of Atg13 mutants were described previously (*Yamamoto et al., 2016*). Plasmids for Atg12 expression were constructed as follows. pRS424-*ATG12* (*Hanada and Ohsumi, 2005*) was digested by *Sac*I and *Xho*I, and a DNA fragment encompassing the *ATG12* gene was ligated with the pRS316 vector (*Sikorski and Hieter, 1989*) cut with the same enzymes, resulting in pRS316-*ATG12*. The nucleotide sequence encoding Ser2 to Gln56 of Atg12 was removed from this plasmid using the QuikChange site-directed mutagenesis kit (Agilent Technologies) to obtain the plasmid expressing Atg12$^{\Delta N56}$ (pRS316-*atg12*$^{\Delta N56}$). DNA fragments excised from these plasmids were ligated with the pRS303 vector (*Sikorski and Hieter, 1989*) in the same manner to construct pRS303-*ATG12* and pRS303-*atg12*$^{\Delta N56}$. These plasmids were digested by *Nhe*I prior to their introduction into yeast cells for integration at the *HIS3* locus. pGAD-*ATG1*, pGAD-*ATG13*, pGAD-*ATG17*, pGAD-*ATG19*, pGBD-*ATG8*, and pGBD-*ATG12* for yeast two-hybrid assay were described previously (*Kabeya et al., 2005*; *Kamada et al., 2000*; *Mizushima et al., 1999*; *Nakatogawa et al., 2012b*; *Noda et al., 2008*). To construct pGAD-*ATG29* and pGAD-*ATG31*, the open reading frames of *ATG29* and *ATG31* were cloned into the *Bam*HI and *Pst*I sites on pGAD-C1 using the Gibson assembly kit (New England Biolabs).

### Fluorescence microscopy

Fluorescence microscopy was performed using an inverted fluorescence microscope (IX83; Olympus) equipped with an electron-multiplying CCD camera (ImagEM C9100-13; Hamamatsu Photonics), and a 150× objective lens (UAPON 150XOTIRF, NA/1.45; Olympus). GFP and mCherry were excited using a 488 nm blue laser (50 mW; Coherent) and a 588 nm yellow laser (50 mW; Coherent), respectively. Fluorescence was filtered with a dichroic mirror reflecting 405 nm, 488 nm, and 588 nm wavelengths (Olympus), separated into two channels using the DV2 multichannel imaging system (Photometrics) equipped with a Di02-R594-25×36 dichroic mirror (Semrock), and further filtered with the TRF59001-EM ET bandpass filter (Chroma) for the GFP channel and the FF01-624/40-25 bandpass filter (Semrock) for the mCherry channel. Images were acquired using MetaMorph software (Molecular Devices) and processed using Fiji (ImageJ) (*Schindelin et al., 2012*; *Schneider et al., 2012*).

**Table 1.** Yeast strains used in this study.

| Name | Genotype | Figures | Reference |
|---|---|---|---|
| W303-1a | MATa ade2-1 ura3-1 his3-11, 15 trp1-1 leu2-3,112 can1-100 | - | (*Thomas and Rothstein, 1989*) |
| ScKH146 | W303-1A, ade2::ADE2 ATG5-EGFP-kanMX6 ATG17-2×mCherry-hphNT1 | 1A | This study |
| ScKH153 | ScKH146 atg16Δ::natNT2 | 1A | This study |
| ScKH182 | ScKH146 atg21Δ::zeoNT3 | 1A | This study |
| ScKH151 | ScKH146 atg14Δ::natNT2 | 1A | This study |
| ScKH149 | ScKH146 atg12Δ::natNT2 | 1A | This study |
| ScKH162 | ScKH146 atg14Δ::natNT2 atg12Δ::zeoNT3 | 1A | This study |
| ScTK623 | ScKH146 atg21Δ::natNT2 atg12Δ::zeoNT3 | 1A | This study |
| BJ2168 | MATa leu2 trp1 ura3-52 prb1-1122 pep4-3 prc1-407 gal2 | 2A-C, 2E, 2F, 2-S1A, 2-S1C | (*Jones, 1991*) |
| MAN169 | BJ2168 ATG5-TEV-3×FLAG-kanMX4 | 2A, 2C, 2E, 2F, 2-S1A, 2-S1C | This study |
| ScKH10 | MAN169 atg16Δ::natNT2 | 2A, 2E, 2-S1A | This study |
| ScKH32 | MAN169 atg12Δ::natNT2 | 2A, 3A | This study |
| ScKH96 | MAN169 atg10Δ::natNT2 | 2A | This study |
| ScKH90 | BJ2168 ATG16-TEV-3×FLAG-kanMX4 | 2B | This study |
| ScKH92 | ScKH90 atg5Δ::natNT2 | 2B | This study |
| ScKH93 | ScKH90 atg12Δ::natNT2 | 2B | This study |
| ScKH141 | MAN169 atg1Δ::natNT2 | 2C | This study |
| ScKH99 | MAN169 atg13Δ::natNT2 | 2C, 2D | This study |
| ScKH216 | MAN169 atg17Δ::natNT2 | 2C | This study |
| ScKH101 | MAN169 atg29Δ::natNT2 | 2C | This study |
| ScKH143 | MAN169 atg31Δ::natNT2 | 2C | This study |
| ScKH98 | MAN169 atg14Δ::natNT2 | 2E | This study |
| ScKH97 | MAN169 atg11Δ::natNT2 | 2F | This study |
| ScYH3184 | BJ2168 leu2::LEU2 | 2G | This study |
| ScKH66 | ScHY3184 ATG5-TEV-3×FLAG-kanMX4 | 2G | This study |
| ScKH68 | ScHY3184 atg1$^{D211A}$-hphNT1 ATG5-TEV-3×FLAG-kanMX4 | 2G | This study |
| AH109 | MATa trp1-901 leu2-3, 112 ura3-52 his3-200 gal4Δ gal80Δ LYS2::GAL1$_{UAS}$-GAL1$_{TATA}$-HIS3 MEL1 GAL2$_{UAS}$-GAL2$_{TATA}$-ADE2 URA3:: MEL1$_{UAS}$-MEL1$_{TATA}$-lacZ | 2-S2A | Clontech |
| ScTK967 | AH109 atg13Δ::natNT2 atg31Δ::hphNT1 | 2-S2B | This study |
| ScTK968 | AH109 atg17Δ::natNT2 | 2-S2C | This study |
| ScTK877 | BJ2168 atg11Δ::LEU2 atg29Δ::zeoNT3 atg31Δ:: hphNT1 atg1Δ::natNT2 atg13Δ::Klura3 P$_{ADH1}$-ATG17-CgTRP1 | 2-S2D | This study |
| ScTK958 | ScTK877 KlURA3-P$_{ADH1}$-ATG12-EGFP-kanMX4 | 2-S2D | This study |

*Table 1 continued on next page*

*Table 1 continued*

| Name | Genotype | Figures | Reference |
|------|----------|---------|-----------|
| ScTK649 | ScTK623 *his3-11*::pRS303 | 3B | This study |
| ScTK650 | ScTK623 *his3-11*::pRS303-ATG12 | 3B | This study |
| ScTK651 | ScTK623 *his3-11*::pRS303-atg12$^{\Delta N56}$ | 3B | This study |
| BY4741 | *MATa his3Δ1 leu2Δ0 met15Δ0 ura3Δ0* | - | (*Brachmann et al., 1998*) |
| ScTK557 | BY4741 *pho8Δ*::*kanMX4-P$_{GPD}$-pho8Δ60 atg12Δ*::*natNT2* | 3C, 4A | This study |
| ScTK559 | BY4741 *pho8Δ*::*kanMX4-P$_{GPD}$-pho8Δ60 atg21Δ*::*zeoNT3 atg12Δ*::*natNT2* | 3C, 4A | This study |
| ScKH119 | W303-1A, *ade2Δ*::*ADE2 ATG17-EGFP-kanMX4 atg11Δ*::*zeoNT3* | 4B | This study |
| ScKH121 | ScKH119 *atg5Δ*::*natNT2* | 4B | This study |
| ScKH123 | ScKH119 *atg12Δ*::*natNT2* | 4B | This study |
| ScKH125 | ScKH119 *atg16Δ*::*natNT2* | 4B | This study |
| ScTK657 | YKH123 *his3-11*::pRS303 | 4C | This study |
| ScTK658 | YKH123 *his3-11*::pRS303-ATG12 | 4C | This study |
| ScTK659 | YKH123 *his3-11*::pRS303-atg12$^{\Delta N56}$ | 4C | This study |

DOI: https://doi.org/10.7554/eLife.43088.010

## Immunoprecipitation

Cells expressing FLAG-tagged proteins were grown to mid-log phase and converted to spheroplasts by incubating them in 0.5× YPD or 0.5× SD+CA+ATU medium containing 1 M sorbitol and 0.1 mg/mL zymolyase 100T (Nacalai tesque) at 30°C for 45 min. These cells were then washed with 20 mM HEPES-KOH (pH7.2) containing 1.2 M sorbitol, and incubated in 0.5× YPD or 0.5× SD+CA+ATU medium containing 1 M sorbitol and 0.2 µg/mL rapamycin at 30°C for 2 hr. The cells were pelleted and solubilized in IP buffer [50 mM Tris-HCl (pH8.0), 150 mM NaCl, 10% glycerol, 5 mM EDTA, 5 mM EGTA, and 50 mM NaF] containing 2 mM phenylmethylsulfonyl fluoride (PMSF), 2× cOmplete Protease Inhibitor Cocktail (Roche), and 1% n-dodecyl-β-maltoside (DDM). After removal of cell debris by centrifugation at 15,000 *g* for 20 min, the resulting supernatants (input) were incubated with NHS FG-beads (Tamagawa Seiki) conjugated with anti-FLAG M2 antibody (F1804; Sigma-Aldrich) or GFP-nanobody (GFP-binding protein) (*Kotani et al., 2018*) and rotated at 4°C for 2 hr. The beads were washed three times with IP buffer containing 0.1% DDM, and bound proteins were eluted with SDS sample buffer [50 mM Tris-HCl (pH7.5), 2% SDS, 8% glycerol, and a trace amount of bromophenol blue] at 65°C for 10 min. Dithiothreitol was added to these samples to a final concentration of 100 mM, and they were incubated at 65°C for 10 min. The samples were analyzed by immunoblotting using antibodies against FLAG (F1804; Sigma), Atg12, Atg5, Atg1, Atg13, Atg17, Atg29, and Atg31 (*Kuma et al., 2002*; *Matsuura et al., 1997*; *Kamada et al., 2000*; *Kawamata et al., 2008*).

## Analysis of Atg8 lipidation

Yeast cells were grown to mid-log phase, treated with 1 mM PMSF for 10 min, and subjected to starvation in SD-N medium containing 1 mM PMSF. Samples for immunoblotting analysis were prepared as described previously (*Kotani et al., 2018*), and subjected to urea-SDS-PAGE (*Nakatogawa and Ohsumi, 2012*) to separate lipidated Atg8 (Atg8-PE) from the unmodified form, followed by immunoblotting using antibodies against Atg8 (anti-Atg8-2) (*Nakatogawa et al., 2012a*).

## Alkaline phosphatase assay

Lysates were prepared from cells before or after starvation for 4 hr in SD-N medium. The ALP assay was carried out as described previously (*Noda et al., 1995*; *Noda and Klionsky, 2008*).

## Acknowledgements

We thank the members of our laboratories for materials, discussions, and technical and secretarial support as well as the Biomaterial Analysis Center, Technical Department at Tokyo Institute of Technology for DNA sequencing service. This work was supported in part by KAKENHI Grants-in-Aid for Scientific Research 25111003 (to HN), 17H01430 (to HN), and 23000015 (to YO) from the Ministry of Education, Culture, Sports, Science, and Technology of Japan, JST CREST Grant Number JPMJCR13M7 (to HN), and STAR Grant funded by the Tokyo Tech Fund (to HN).

## Additional information

### Competing interests

Hitoshi Nakatogawa: Reviewing editor, *eLife*. The other authors declare that no competing interests exist.

### Funding

| Funder | Grant reference number | Author |
| --- | --- | --- |
| Ministry of Education, Culture, Sports, Science, and Technology | 25111003 | Hitoshi Nakatogawa |
| Ministry of Education, Culture, Sports, Science, and Technology | 17H01430 | Hitoshi Nakatogawa |
| Ministry of Education, Culture, Sports, Science, and Technology | 23000015 | Yoshinori Ohsumi |
| Japan Science and Technology Agency | JPMJCR13M7 | Hitoshi Nakatogawa |

The funders had no role in study design, data collection and interpretation, or the decision to submit the work for publication.

### Author contributions

Kumi Harada, Conceptualization, Data curation, Formal analysis, Validation, Writing—original draft, Writing—review and editing; Tetsuya Kotani, Data curation, Formal analysis, Validation, Writing—review and editing; Hiromi Kirisako, Machiko Sakoh-Nakatogawa, Yayoi Kimura, Hisashi Hirano, Data curation, Validation, Writing—review and editing; Yu Oikawa, Data curation, Writing—review and editing; Hayashi Yamamoto, Resources, Validation, Writing—review and editing; Yoshinori Ohsumi, Conceptualization, Funding acquisition, Validation, Writing—review and editing; Hitoshi Nakatogawa, Conceptualization, Supervision, Funding acquisition, Validation, Writing—original draft, Project administration, Writing—review and editing

### Author ORCIDs

Hitoshi Nakatogawa (iD) http://orcid.org/0000-0002-5828-0741

### Decision letter and Author response

Decision letter https://doi.org/10.7554/eLife.43088.013
Author response https://doi.org/10.7554/eLife.43088.014

# Additional files

## Supplementary files
• Transparent reporting form
DOI: https://doi.org/10.7554/eLife.43088.011

## Data availability
All data generated or analysed during this study are included in the manuscript and supporting files.

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
