## [Decision Letter]

Thank you for submitting your article "Two distinct mechanisms target the autophagy-related E3 complex to the pre-autophagosomal structure" for consideration by *eLife*. Your article has been reviewed by three peer reviewers, one of whom is a member of our Board of Reviewing Editors, and the evaluation has been overseen by Ivan Dikic as the Senior Editor. The reviewers have opted to remain anonymous.

The reviewers have discussed the reviews with one another and the Reviewing Editor has drafted this decision to help you prepare a revised submission.

Summary:

Upon autophagy induction in yeast, a set of Atg proteins is recruited in a hierarchical order to the PAS. The Atg1 complex acts in the most upstream of the recruitment. Previous studies have shown that the Atg12-Atg5-Atg16 complex, which acts as a E3 for promoting lipidation of Atg8, is targeted to PAS via interaction with Atg21 in a PI(3)P-dependent manner. In this manuscript, the authors revealed a novel mechanism for the PAS targeting of the Atg12-Atg5-Atg16 complex. They demonstrated that this complex can be recruited to the PAS via direct interaction of the N-terminal region of Atg12 with the Atg1 complex. The formation of Atg17 puncta is reduced in the absence of the Atg12-Atg5-Atg16 complex, suggesting a crosstalk between these two complexes. These findings provide novel and useful information for our understanding of the early stage of autophagy process. The experiments are well-designed, the data are with high quality, and the manuscript is well-written. It is suitable for publication in *eLife* after appropriate revisions.

Essential revisions:

1) Which component(s) in the Atg1 complex interact with the N-terminal 56 amino acids of Atg12?

2) Does the N-terminal part of Atg12 have the ability to localize by themselves to the PAS? The authors should try if proteins (such as GFP) can be directed to the PAS via this domain.

---

## [Author Response]

Essential revisions:1) Which component(s) in the Atg1 complex interact with the N-terminal 56 amino acids of Atg12?

Thank you for asking this important question. In the revised manuscript, we first performed yeast two-hybrid assay to examine the interactions of all the Atg1 complex components with Atg12, and consequently, only Atg17 was suggested to directly interact with Atg12 (Figure 2—figure supplement 2A-C). We further confirmed this interaction by coimmunoprecipitation analysis in cells lacking the four components of the Atg1 complex other than Atg17 (Figure 2—figure supplement 2D). Thus, these new results suggest that Atg17 is the component of the Atg1 complex that interacts with Atg12.

2) Does the N-terminal part of Atg12 have the ability to localize by themselves to the PAS? The authors should try if proteins (such as GFP) can be directed to the PAS via this domain.

Given that Atg5-GFP, which should exist as a conjugate with Atg12, cannot localize to the PAS in the absence of Atg16, the N-terminal region of Atg12 alone was thought not to localize to the PAS, either. However, as this was an interesting open question, we experimentally examined it and showed that Atg12^WT^-GFP, Atg12^N56^-GFP,and Atg12^N100^-GFPwere dispersed throughout the cytoplasm and unable to localize to the PAS as shown below (Author response image 1).

**Author response image 1. respfig1:** Cells were treated with rapamycin for 2 h, and examined by fluorescence microscopy.

We did not include these results in the revised manuscript.